



# Atmospheric NH₃ in urban Beijing: long-term variations and implications for secondary inorganic aerosol control

Ziru Lan[1], Xiaoyi Zhang[2], Weili Lin[1], Xiaobin Xu[2], Zhiqiang Ma[3], Jun Jin[1], Lingyan Wu[2], Yangmei Zhang[2]

[1]Key Laboratory of Ecology and Environment in Minority Areas, Minzu University of China, National Ethnic Affairs Commission, Beijing 100081, China
[2]Institute of Atmospheric Composition, Chinese Academy of Meteorological Science, Beijing 100081, China
[3]Institute of Urban Meteorology, China Meteorological Administration, Beijing 100089, China

*Correspondence to*: Weili Lin(linwl@muc.edu.cn)

**Abstract**

Ammonia (NH₃) has major effects on the environment and climate. In-situ measurements of NH₃ concentrations taken between June 2009 and July 2020 at an urban site in Beijing were analyzed to study their long-term behaviors, responses to meteorological conditions and influences on the formation of secondary inorganic aerosols (SIAs). The total average NH₃ mixing ratio was $26.9 \pm 19.3$ ppb (median, 23.5 ppb). NH₃ mixing ratios initially increased and peaked in 2017 but

subsequently decreased, resulting in an overall decrease of 24% from 2009 to 2020. Notably, the long-term trend for NH₃ at the ground level did not align with the trends derived from satellite observations and emission estimates. It exhibited distinct seasonal variation but also complex diurnal patterns across multiple seasons and years. The NH₃ concentration exhibited a stronger correlation with the water vapor (H₂O) concentration than with air temperature. Thermodynamic modeling revealed the nonlinear response of SIAs to NH₃. Although reducing NH₃ concentrations can improve air quality during winter,

controlling acid gas concentrations has a greater effect than controlling NH₃ concentrations on reducing SIA concentrations. The increase in the proportion (mass concentration) of ammonium salts in SIAs during the observation period indicates that measures to control NH₃ concentrations should be prioritized.

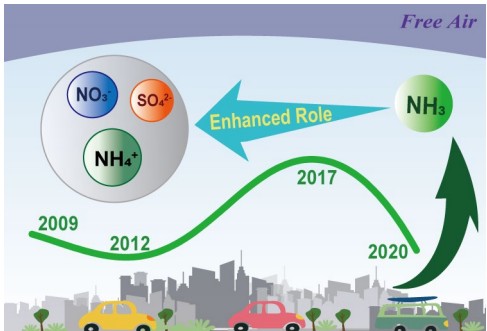





# 1 Introduction

In the atmosphere, NH₃ is a major reduced nitrogen compound that has considerable effects on ecosystem nitrogen cycling, climate change, atmospheric particulate matter, and acid rain formation. As the most abundant alkaline trace gas in the atmosphere(Meng et al., 2017), NH₃ interacts with the oxidized products of atmospheric acidic gases to form secondary aerosols, which considerably affect the radiative balance of the atmosphere and air quality (Fuzzi et al., 2015). On one hand, NH₃ reduces the acidity of precipitation by neutralizing acidic substances. On the other hand, the ammonium ions ($NH_4^+$)

formed in precipitation can cause severe soil acidification when microbial processes are involved. The natural cycling of NH₃ and its transformation products alone are not necessarily harmful. However, the Haber–Bosch process has led to a rapid increase in synthetic NH₃ production (Erisman et al., 2008, 2007; Fowler et al., 2015). In addition, large-scale livestock farming and other activities have contributed considerably to NH₃ emissions. The excessive input of anthropogenic nitrogen into the environment can directly harm ecosystems and influence climate change(Charlson et al., 1991; Reay et al., 2008;

Shadman et al., 2016).

A growing body of evidence regarding atmospheric NH₃ has highlighted the importance of reducing NH₃ emissions. After its commitment to controlling air pollution for several years, China has managed to effectively control the emissions of primary pollutants such as sulfur dioxide ($SO_2$) and nitrogen oxide ($NO_x$). However, particulate matter 2.5 ($PM_{2.5}$) pollution is still a severe problem. Research on controlling $SO_2$ and $NO_x$ emissions indicate that controlling NH₃ emissions is the most

economically effective way for reducing $PM_{2.5}$ concentrations (Gu et al., 2021; Pinder et al., 2008). After a comprehensive review, Xie et al. (2022) suggested that the most effective strategy for mitigating wintertime nitrate pollution in North China is to reduce NH₃ emissions. The signing of the 1999 United Nations Economic Commission for Europe Gothenburg Protocol marked the first time that NH₃ was included in the air pollutant emission control system(Reis et al., 2012). In 2018, the State Council of China added NH₃ emission reduction as an air pollution control objective and, for the first time, emphasized the

need to control agricultural NH₃ emissions (The State Council of China, 2018). In December 2021, the *14th Five-Year Plan for Ecological and Environmental Protection in Beijing* was announced, and it was the first 5-year plan to highlight the importance of controlling atmospheric NH₃ emissions and to set clear emission reduction targets for NH₃(The People's Government of Beijing Municipality, 2021).

Long-term observations are important for analyzing the environmental impacts and control strategies of atmospheric NH₃.

Countries such as the United Kingdom (Sutton et al., 2001; Tang et al., 2018), the Netherlands (Lolkema et al., 2015), and the United States (Butler et al., 2016) have established NH₃ monitoring networks and have conducted long-term observational studies spanning more than a decade. In addition to national monitoring networks, specific regions in Hungary (Horvath et al., 2009; Horváth and Sutton, 1998), Japan (Yamamoto et al., 1995, 1988), Belgium (den Bril et al., 2011), Canada (Yamanouchi et al., 2021), and India (Saraswati et al., 2017) have conducted studies on NH₃ variations over a period

of 5 years or more. In most of these regions, NH₃ concentrations have either remained stable or have exhibited an increasing



trend. In China, studies of long-term variations in atmospheric NH$_3$ concentrations showed significant increasing trends. The Nationwide Nitrogen Deposition Monitoring Network (NNDMN) began to monitor gaseous NH$_3$ in 2010 with monthly temporal resolution (Xu et al., 2019). Based on the monitoring results from 66 stations of the NNDMN, NH$_3$ concentrations increased throughout China from 2011 to 2018(Wen et al., 2020). Luo et al. (2020) also found a rapid increase in NH$_3$

concentration from 2011 to 2018 in North China. Zhou et al. monitored atmospheric NH$_3$ with an online analyzer in East China from 2011 to 2019 but did not analyze NH$_3$ variations (Zhou et al., 2022). In addition to ground-based measurements, satellite observations enabled the retrieval of NH$_3$ in the atmosphere with high spatiotemporal resolution (Clarisse et al., 2021; Liu et al., 2022). Dong et al. (2023) analyzed satellite data and suggested a significant increase (~32%) in NH$_3$ vertical column densities in China from 2008 to 2019. To effectively utilize satellite data, it is important to validate these

observations with in-situ surface measurements (Pinder et al., 2011; Van Damme et al., 2015; Van Damme et al., 2021). However, long-term ground-based observations of atmospheric NH$_3$ at high temporal resolution are relatively rare, and studies focusing on the long-term trends of atmospheric NH$_3$ in urban areas of China are lacking.

The present study examined high temporal resolution NH$_3$ observations at the surface in urban Beijing from 2009 to 2020. Using data from emission inventories, satellite observations, meteorological elements, concentrations of various types of

atmospheric pollutants, and particle ion composition, the present study aims to obtain the characteristics of long-term variations, influencing factors, and the contributions of NH$_3$ to particle formation in the atmosphere of Beijing. Analyzing long-term NH$_3$ observations can help to understand how changes in NH$_3$ concentrations have affected atmospheric pollution in the past. This knowledge is crucial for predicting future atmospheric pollution and formulating effective environmental policies. Additionally, it provides a scientific basis and reference for developing future NH$_3$ control strategies.

## 2 Materials and methods

### 2.1 Data

Between June 2009 and July 2020, data on continuous online measurements of NH$_3$ concentrations were collected in Haidian District, Beijing (39°95'N, 116°32'E, Figure S1). From June 2009 to September 2017, data were collected from an observational site located on the third floor of a building within the premises of the China Meteorological Administration.

Subsequently, the observation site was relocated to the 14th floor of the Science and Technology Building of Minzu University of China, which was less than 1 km away from the previous location and located just across the road from it. Both observation sites were surrounded mostly by urban roads, office spaces, residential areas, and parks, and no large-scale industrial sources of NH$_3$ were located near the site.

Beginning in June 2009, NH$_3$ concentration monitoring was conducted using an EC9842 NO$_x$/NH$_3$ Analyzer (Ecotech,

Australia). The air had been drained into an air-conditioned room with a 4.5 m long Teflon line and the inlet height is 1.8 m



above the rooftop (about 12 m above ground level). Starting in April 2015, additional $NH_3$ measurements were simultaneously taken using an EAA $NH_3$ Analyzer (Los Gatos Research, USA). From May 2016 onward, only the EAA $NH_3$ Analyzer was used. The EC9842 $NO_x$/$NH_3$ Analyzer employs gas-phase chemiluminescence to continuously analyze $NH_3$, $NO_x$, and $N_x$ concentrations, its detection limit is less than 2 ppb and data record time is 1 minute. The instrument was subjected to weekly zero and span checks to identify potential analyzer faults and response drift. Multipoint calibrations were typically performed every month, and data were corrected on the basis of the multipoint calibrations. The EAA $NH_3$ Analyzer features a low detection limit of less than 0.2 ppb and a maximum drift of 0.2 ppb within 24 hours, with a time resolution of 50 seconds, and it utilizes Off-Axis Integrated Cavity Output Spectroscopy technology. To maintain data comparability, $NH_3$ standard gases, which had been traceable to a uniform standard, were used as measurement references. The comparison result of the two instruments can be found in Zhang et al. (2021), in which the two instruments exhibited a considerable correlation, with a correlation coefficient of 0.949 (n = 5316, p < 0.01) and slope of s 0.999 ± 0.005.

During data analysis, minute-level data were converted into hourly average data. Throughout the observation period, a total of 40,692 and 46,917 valid hourly average data points were obtained from the EC9842 and EAA analyzers, respectively, resulting in a total of 13,420 data sets being obtained simultaneously through measurements on the two instruments. These two sets of results exhibited a significant correlation (N = 13,420, slope = 1.09, R = 0.95, p < 0.05), and the parallel observations from the two analyzers were generally consistent (Figure S2). The $NH_3$ observation data were finalized by averaging the synchronized data.

Furthermore, $NH_3$ satellite observation data were obtained through Metop-A satellite's Infrared Atmospheric Sounding Interferometer (IASI) remote sensing product. These data had a spatial resolution of $12 \times 12$ km$^2$ and were collected on a monthly basis (Van Damme et al., 2017). In the present study, daytime satellite $NH_3$ data from June 2009 to April 2020 were used. The average $NH_3$ satellite observation results for Beijing were calculated using data for the region spanning 36.5°N to 42.5°N in latitude and 113.5°E to 118.5°E in longitude. The trend for satellite observation values obtained at the grid point at the location of the monitoring station closely matched the trend for the average observation values collected for this region (Figure S3). $NH_3$ emission inventory data for Beijing (from June 2009 to December 2017) were presented in Figure S4, comparing $NH_3$ emissions from Beijing and its surrounding areas(Huang et al., 2012; Kang et al., 2016). Meteorological data collected between June 2009 and February 2012 were obtained from the Beijing Capital International Airport station. From March 2012 to April 2020, meteorological data were sourced from the Haidian Meteorological Station. The temperature and relative humidity data acquired from the two stations exhibited a high level of correlation (Figure S5). Absolute humidity was calculated using the acquired temperature and relative humidity data. Data for other pollutants such as $PM_{2.5}$, $SO_2$, and $NO_2$ were acquired from the Wanliu Monitoring Station in Haidian District, Beijing. These monitoring data were collected between April 2, 2014, and July 11, 2020. Figure S6 provides additional details of these data.



In the present study, offline sampling of $PM_{2.5}$ was conducted on the rooftop of the School of Pharmacy at Minzu University in China. Atmospheric samples were collected twice daily, specifically from 6:00 to 17:00 (daytime sampling) and from 18:00 to 5:00 on the following day (nighttime sampling). The sampling periods were from September 8 to 21, 2018 (autumn);

November 6 to 21, 2018 (autumn); January 1 to 21, 2019 (winter); March 3 to 21, 2019 (spring); May 8 to 15, 2019 (spring); and June 8 to 21, 2019 (summer). $PM_{2.5}$ samples collected on filters were analyzed for ion components ($Na^+$, $SO_4^{2-}$, $NH_4^+$, $NO_3^-$, $Cl^-$, $Ca^{2+}$, $K^+$, and $Mg^{2+}$) at the Chinese Academy of Meteorological Sciences, resulting in the acquisition of 184 data sets. Additionally, data from the study by Hu et al. (2014) that spanned from May 5 to November 30, 2009, and data from the study of Wu et al. (2019) that spanned December 15 to 23, 2016, were used in the present study as references for

monitoring $PM_{2.5}$ components within the premises of the China Meteorological Administration.

## 2.1 Methods

### 2.2.1 Long-term trends analysis

Long-term trends of atmospheric $NH_3$ were obtained using Ensemble Empirical Mode Decomposition (EEMD)(Wu and Huang, 2009). This method adaptively decomposes a signal into a series of Intrinsic Modal Functions (IMFs) from high to

low frequencies. It separates oscillation or trend components of varying scales from the original signal. EEMD integrates the advantages of wavelet analysis and augments the Empirical Mode Decomposition (EMD) method by introducing white noise. This enhancement effectively mitigates the mode mixing problem inherent in the EMD method. EEMD demonstrates greater stability in decomposing nonlinear and non-stationary data series, enabling the accurate extraction of genuine signal variations(Qian et al., 2011). Currently, EEMD has been used in studies on climate change and pollutant forecasting (Fu et

al., 2020; Ji et al., 2014; Lee and Ouarda, 2011; Xu et al., 2022). In the present study, the EEMD was performed using the Rlibeemd package of the R programming language (Luukko et al., 2016).

### 2.2.2 Thermodynamic modeling

To assess the sensitivity of sulfate, nitrate, and ammonium (SNA) to changes in precursor concentrations, the present study employed the ISORROPIA II thermodynamic equilibrium model in the forward-problem calculation mode. The model was

used to analyze the sensitivity of SNA to changes in precursor concentrations under "metastable stable" state conditions because of its high precision and computational efficiency, which have been widely demonstrated(Fountoukis and Nenes, 2007). The ISORROPIA-II model is mainly used to simulate the physical state and concentration of inorganic components of the aerosol system at thermodynamic equilibrium. A distinct advantage of the ISORROPIA-II model over other thermodynamic models is the inclusion of the $K^+$, $Ca^{2+}$, and $Mg^{2+}$ ions in the calculations, and taking these components into

account significantly improves the accuracy of the model simulations (Allen et al., 2015).



## 3 Results and discussion

### 3.1 Long-term variations in NH₃

From June 2009 to July 2020, the hourly average mixing ratio of atmospheric NH$_3$ in Beijing was 26.9 ± 19.3 ppb (median, 23.5 ppb). Table S1 summarizes results from various NH$_3$ monitoring studies conducted in urban areas. The results of the present study are basically consistent with the annual NH$_3$ mixing ratio averages that were observed in urban Beijing by other researchers through optical instruments during the present study period(Gu et al., 2022a, 2022b; Pu et al., 2020; Sun et al., 2023; Wang et al., 2019). However, the concentration values obtained through optical instruments are notably higher than the NH$_3$ mixing ratios measured using chemical absorption methods (Meng et al., 2011; Pan et al., 2018; Su et al., 2021). Apart from differences in monitoring locations and time periods, differences in the types of instruments used can affect the monitoring results. Von Bobrutzki et al. reported that an acoustic instrument overestimated NH$_3$ concentrations (von Bobrutzki et al., 2010). Stieger et al. (2017) compared the performance of MARGA and Picarro instruments in low-concentration environments and reported that Picarro instruments recorded higher NH$_3$ measurements. Twigg et al. (2022)conducted a comprehensive comparison of 13 NH$_3$ monitoring instruments and discovered that these instruments obtained similar values at higher NH$_3$ concentrations but exhibited larger differences at lower concentrations.

As densely populated country with intensive agriculture activities, China contains several areas that are major global hotspots for the atmospheric NH$_3$ concentration (Liu et al., 2019; Van Damme et al., 2018). The monitoring results of the present study indicate that the overall NH$_3$ mixing ratio in Beijing is lower than that in Delhi(Saraswati et al., 2019; Singh and Kulshrestha, 2014) but considerably higher than those in other developed cities such as New York, Toronto, and Rome(Chatain et al., 2022; Nguyen et al., 2021; Park et al., 2021; Perrino et al., 2002; Phan et al., 2013; Zbieranowski and Aherne, 2012; Zhou et al., 2019). Even within China, the NH$_3$ mixing ratio in Beijing is higher relative to that in Shanghai, which is also a megacity (i.e., the NH$_3$ mixing ratio in Shanghai is less than one-third of that in Beijing), and only a few cities in North China have mixing ratios comparable to that in Beijing(Cao et al., 2009; Chang et al., 2019; Huang et al., 2021; Pan et al., 2018). The primary reasons for this phenomenon are the frequent agricultural activities and the presence of highly alkaline soils in the North China Plain, where Beijing is located(Ju et al., 2009). Over the past 2 decades, Beijing has implemented a series of strict measures to control air pollution and has achieved considerable success(United Nations Environment Programme, 2019). The concentrations of SO$_2$, NO$_2$, CO, PM$_{10}$, and PM$_{2.5}$ in Beijing all exhibited decreasing trends; in particular, the concentration of SO$_2$ decreased by 88% from 2009 to 2020. To analyze the long-term trends of the atmospheric NH$_3$ concentration, the present study referred to the findings of Vu et al. (2019) and used meteorological factors to construct a random forest model for imputing missing values. The computed time series for the atmospheric NH$_3$ concentration is presented in Figure S7. Unlike those of other primary pollutants, the annual average concentration for NH$_3$ exhibited a general decreasing trend but an initial increase followed by a decline. The annual average NH$_3$ concentration in Beijing peaked in 2017, however, the annual average NH$_3$ concentration in 2020 was 24% lower than that in 2009 (Figure 1).



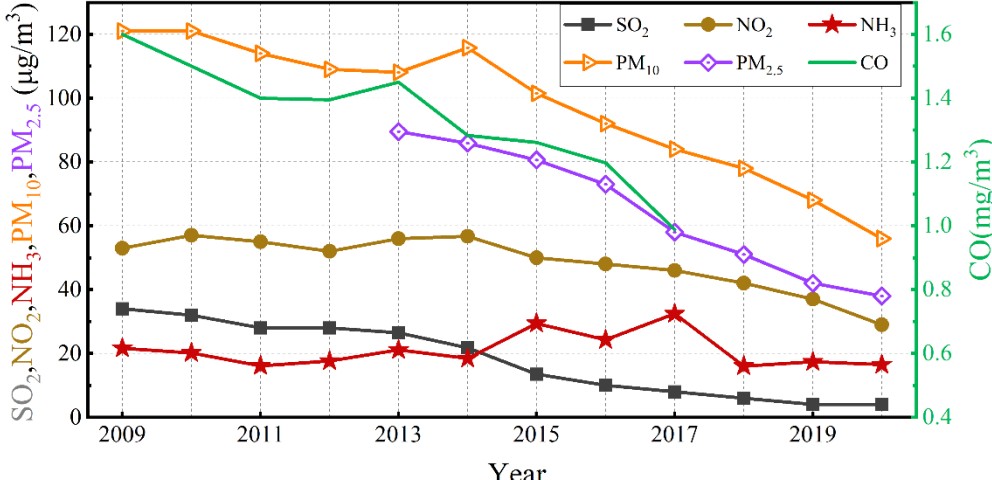

**Figure 1: Annual average concentrations of atmospheric NH₃ and six air pollutants in Beijing. The measurement unit is mg/m³ for CO concentration and μg/m³ for all other pollutants (air pollutant data were retrieved from the Beijing Environmental Bulletin website: http://sthjj.beijing.gov.cn/bjhrb/index/xxgk69/sthjlyzwg/1718880/1718881/1718882/).**

To compare with satellite monitoring data, the Beijing NH₃ emission inventory and long-term trends obtained through EEMD were used to further characterize the changes in atmospheric NH₃ concentrations in Beijing (Figure 2). A comparison of the monthly average NH₃ concentrations obtained from satellite observations revealed that before 2018, the trend for the surface NH₃ mixing ratio was similar to that observed by satellites, exhibiting a decline followed by an increase in the atmospheric NH₃ concentration. However, starting in 2018, the two trends diverged, with the satellite observations indicating a continued increase in the NH₃ concentration, and the surface NH₃ mixing ratio exhibiting a decreasing trend. Studies have indicated strong agreement between satellite data and ground-based monitoring results(Chen et al., 2020; Van Damme et al., 2015; Wang et al., 2022). The difference identified in the present study could be due to the change in the monitoring location and observation height in September 2017. However, other studies examining the NH₃ mixing ratio in Beijing have reported only slight variations within an altitude range of 300 m(Wang et al., 2019; Zhang et al., 2019). Therefore, the change in observation altitude was unlikely to be the primary reason for the rapid decrease in the surface NH₃ mixing ratio after 2017. Zhang et al. (2020) measured NH₃ concentrations at five stations in Beijing, and four of them provided lower NH₃ concentrations in 2017 (winter) than in 2020 (winter + spring), whereas one station had higher concentrations in 2017 than in 2020. This finding indicates that observation results from different locations can vary, even within the same city. Because of the short atmospheric lifetime, low transport altitude, high dry deposition rate, limited transport distance, and abundance of atmospheric NH₃, its complex temporal and spatial characteristics contribute to the complexity of NH₃ variations(Asman and van Jaarsveld, 1992; Nair and Yu, 2020). Satellite observations are limited by the observation height and spatial resolution, which may mask variations in local surface NH₃ concentrations.





The acquired emission inventory data revealed that prior to 2014, the total NH$_3$ emissions in Beijing remained stable, peaking in 2012. After 2014, NH$_3$ emissions in Beijing rapidly decreased, declining by 25% from 2012 to 2017 and by 18% from 2016 to 2017. However, during this period of declining emissions, the NH$_3$ mixing ratio in Beijing exhibited an increasing trend. Similar phenomena have been reported by studies conducted outside of China. For instance, in Scotland,
NH$_3$ emissions decreased by approximately 15% from 1990 to 2003, whereas atmospheric NH$_3$ concentrations increased (Friedman and Schwartz, 2011). In Hungary, NH$_3$ emissions were estimated to have decreased by 50% from 1983 to 1993; however, NH$_3$ concentrations exhibited a slight upward trend during this monitoring period(Horvath et al., 2009). A possible reason for these differences between NH$_3$ emissions and concentrations could be the significant reduction in the concentrations of SO$_2$ and NO$_x$, which reduced the amount of atmospheric NH$_3$ neutralized by acid gases (Fu et al., 2017;
Lachatre et al., 2019; Liu et al., 2018; Yu et al., 2018), thereby reducing the effectiveness of SO$_2$ and NO$_x$ emission control measures in mitigating PM$_{2.5}$ pollution. A study conducted in North China during winter revealed that under high-NH$_3$ emission conditions, even substantial reductions in NO$_x$ emissions led to an increase in the nitrate content of PM$_{2.5}$(Zhai et al., 2021).

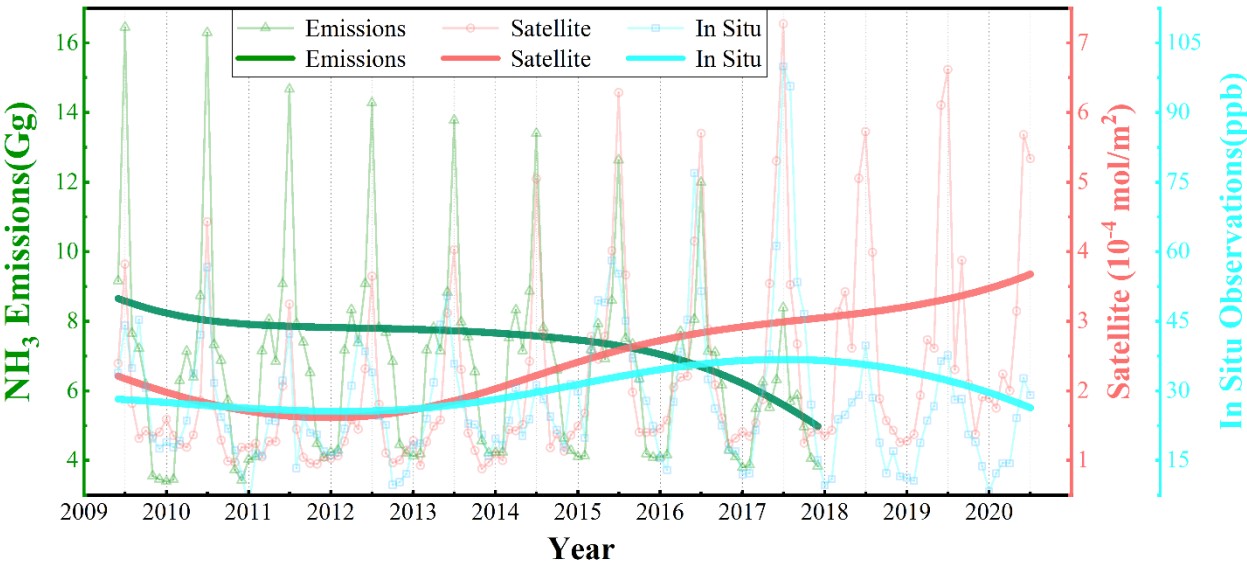

**Figure 2: Monthly averages of surface observations and satellite inversions of NH$_3$ concentrations and total NH$_3$ emissions in Beijing from June 2009 to July 2020 (fine dotted line) and trends pertaining to changes (thick solid line).**

## 3.2 Influence of meteorological elements on NH$_3$

Various meteorological factors can influence the atmospheric NH$_3$ concentration. Among the identified factors, temperature
has been reported to be positively correlated with the NH$_3$ concentration. An increase in temperature can increase soil NH$_3$



emissions, leading to the equilibrium shift of particulate $NH_4NO_3$ toward gaseous $NH_3$, which increases the $NH_3$ concentration(Behera et al., 2013; Li et al., 2014). During the observation period, the temperature in Beijing followed the seasonal sequence of summer (being the warmest), followed by spring, autumn, and winter, and the rankings of $NH_3$ mixing ratios across the seasons were consistent with the trend. Other studies conducted in temperate regions of the Northern

Hemisphere have reported similar findings (Liu et al., 2021; Shon et al., 2012; Wang et al., 2018). The interannual trends pertaining to temperature and $NH_3$ mixing ratios across multiple seasons (Figure 3) revealed that temperature remained stable in summer and autumn over the years; when calculated in Kelvin, the average annual temperature exhibited variation coefficients of 0.42% in spring, 0.15% in summer, 0.17% in autumn, and 0.51% in winter. For the two seasons of summer and autumn, no significant correlation was identified between the annual average $NH_3$ mixing ratio and the variations in

temperature over the years. After 2014, the annual average temperature in spring remained stable, whereas the $NH_3$ mixing ratio gradually decreased, possibly because of a reduction in agricultural activities. A weak positive correlation was identified between the annual average $NH_3$ mixing ratio and temperature only in winter, and the significant increase in winter temperature from 2013 to 2014 could have led to the high $NH_3$ mixing ratios in the winter of 2014.

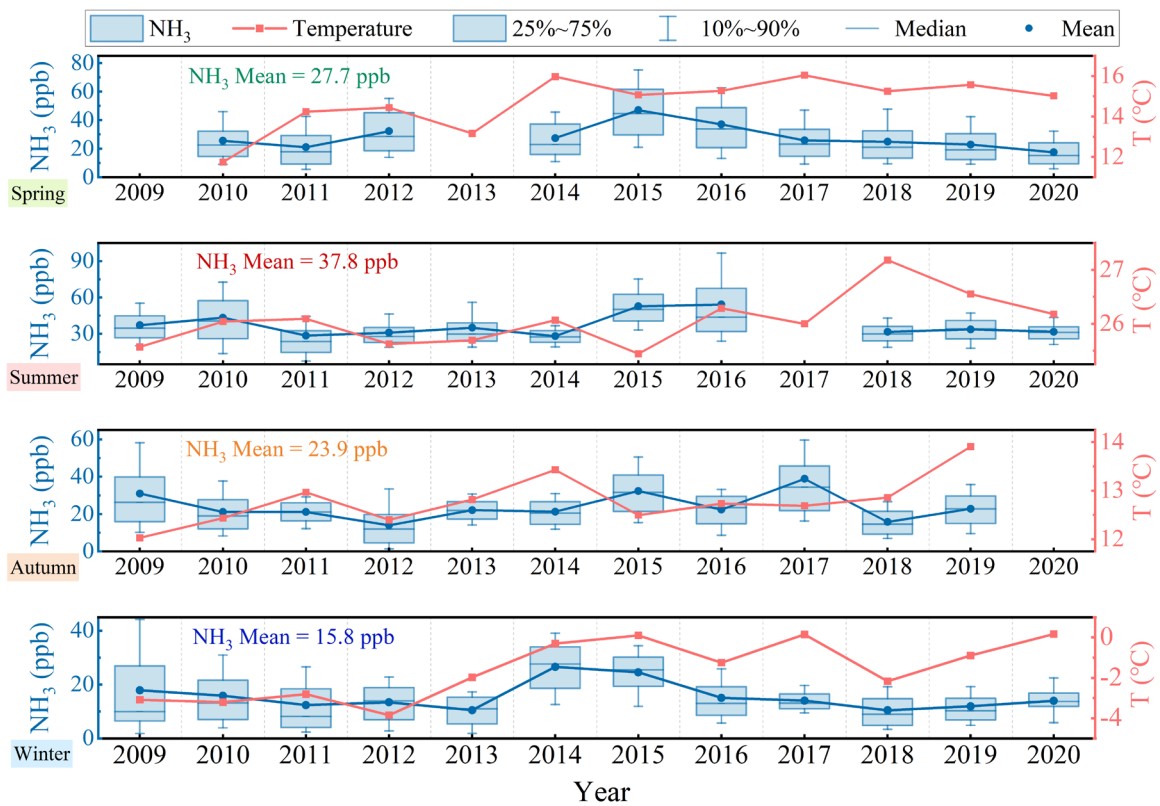

**Figure 3: Interannual variations in mean $NH_3$ mixing ratio and mean temperature in Beijing for each season.**





Several studies have suggested that temperature plays a pivotal role in driving diurnal variations in atmospheric $NH_3$ concentrations (Clarisse et al., 2021; Langford et al., 1992). Our investigation examined the correlations among daily temperature, absolute humidity, and diurnal fluctuations of atmospheric $NH_3$ concentrations throughout an observation period of 4,058 days (screening criteria for effective dates: $p < 0.05$ and $\geq 18$ effective hours per day, the p-values were adjusted using the Benjamini-Hochberg method(Benjamini and Hochberg, 1995)). We observed that temperature exhibited both positive (45%) and negative (55%) correlations with the $NH_3$ mixing ratio, with these two categories each accounting for nearly half of the valid observation days. However, on most days (i.e., 93% of the valid observation days), absolute humidity was positively correlated with the $NH_3$ mixing ratio (Figure S8). Overall, the average daily variations in $NH_3$ mixing ratios in Beijing in spring, summer, and autumn indicated a significant negative correlation with the temperature ($R_{spring} = -0.93$, $R_{summer} = -0.72$, $R_{autumn} = -0.76$, $p < 0.01$). The $NH_3$ mixing ratio was positively correlated with absolute humidity ($R_{spring} = 0.80$, $R_{summer} = 0.50$, $R_{autumn} = 0.67$, $R_{winter} = 0.49$, $p < 0.01$). Several studies reported a high correlation between the $NH_3$ mixing ratio and humidity. Previous research has shown that $NH_3$ can be significantly affected only by sharp changes in humidity, and a new balance requires tens of minutes to be re-established. Averaging minute-level data over one hour can smooth the effects the effect caused by variations in humidity. Notably, parallel observations in urban and suburban Beijing found that a positive correlation between daily $NH_3$ and $H_2O$ concentration variations was only significant in urban areas(Lan et al., 2021). Gu et al. (2022b) reported daily variations in $NH_3$ concentrations in urban Beijing, as measured by the Picarro Ammonia Analyzer and ChemComb were consistent. These suggest that atmospheric $NH_3$ cannot be explained solely by the influence of $H_2O$ effects on the instruments. Additionally, Sun et al. discovered a positive correlation between atmospheric $NH_3$ concentrations and relative humidity (RH) in Beijing and a negative correlation in Shanghai (Sun et al., 2023). In rural North China, He et al. (2020) also observed a strong correlation between $NH_3$ and RH, which they attributed to dew evaporation. At present, the relationship between the $NH_3$ mixing ratio and the water vapor concentration requires further clarification.

In summary, the results infer that temperature plays a pivotal role in driving the seasonal variations in atmospheric $NH_3$ concentrations throughout a given year. However, in the long term, the influence of temperature and other meteorological factors may be masked. Regarding diurnal variations, our analysis revealed that a single-day increase in temperature did not consistently lead to a direct elevation in atmospheric $NH_3$ mixing ratios on most days. Conversely, atmospheric water vapor mixing ratios exhibited a consistently positive correlation with $NH_3$ mixing ratios throughout the day. Notably, the day-to-day variations in meteorological factors remained consistent across the years, whereas the variation in diurnal $NH_3$ differed across different years and seasons (Figure S9). Therefore, the conclusion is that diurnal fluctuations in the atmospheric $NH_3$ concentration are not solely determined by meteorological factors. In recent years, scholars have been increasingly studying the contribution of traffic sources to urban $NH_3$ concentrations. Gu et al. (2022b) confirmed that vehicle exhaust emissions during winter in Beijing lead to the occurrence of morning peaks in $NH_3$ concentrations. Nonetheless, our results indicated that atmospheric $NH_3$ concentrations did not consistently peak in the morning throughout the observation period, even when





high concentrations of traffic emissions were present. Furthermore, the morning peaks for atmospheric NH$_3$ concentrations
tended to occur earlier relative to those for CO, which are influenced by traffic emissions (Figure S9).

Studies have comprehensively explored the influence of wind direction and wind speed on the pollutant mixing ratios in
Beijing, and they have reported that southerly (S) winds transport a high concentration of pollutants to Beijing, leading to the
accumulation of NH$_3$ in the city. Conversely, the winds from the north by west (NW) facilitate the dispersion and dilution of
atmospheric NH$_3$ in Beijing (Lin et al., 2011; Meng et al., 2017). Figure S10 presents the wind rose diagrams for
atmospheric NH$_3$ concentrations in various seasons and wind speeds. Under near-calm wind conditions (wind speed [ws] ≤
1.5 m/s), the prevailing winds across all seasons were predominantly in the northeast and east-northeast directions, and NH$_3$
mixing ratios did not vary significantly because of the wind direction, indicating that local emissions had the most
pronounced effect on atmospheric NH$_3$ concentrations. At low wind speeds (1.6 m/s ≤ ws ≤ 3.3 m/s), the predominant wind
direction varied across seasons, with southwesterly winds prevailing in spring and summer and northerly winds dominating
in autumn and winter. At higher wind speeds (ws > 3.4 m/s), the predominant wind direction was NW in spring, autumn, and
winter but southerly in summer. In general, changes in the prevailing wind direction did not significantly influence NH$_3$
mixing ratios across various wind sectors. However, in specific wind sectors, such as the west by south (WS) sector in spring,
the east by south (ES) sector in summer and autumn, and the south by east (SE) sector in winter, higher wind speeds tended
to lead to lower NH$_3$ mixing ratios (Figure S11). Notably, the decline in NH$_3$ mixing ratios was more pronounced in wind
sectors affected by NW winds, indicating that strong winds, particularly those from the NW direction, had a significant
cleansing effect on NH$_3$ in Beijing. Conversely, southerly winds, and sometimes specific wind directions, contributed to NH$_3$
accumulation.

### 3.3 Influence of NH$_3$ on secondary inorganic aerosol formation

NH$_3$, a primary alkaline gas in the atmosphere, can react with acidic substances. In the atmosphere, NH$_3$ preferentially reacts
with the oxidation products of SO$_2$ to form stable compounds such as (NH$_4$)HSO$_4$ or (NH$_4$)$_2$SO$_4$. When NH$_3$ completely
neutralizes H$_2$SO$_4$, the resulting sulfate primarily exists as (NH$_4$)$_2$SO$_4$. Excess NH$_3$ continues to react with HNO$_3$ and HCl,
leading to the formation of unstable compounds such as NH$_4$NO$_3$ and NH$_4$Cl. These reactions increase the concentrations of
secondary aerosols in the atmosphere (Shon et al., 2012). Secondary inorganic aerosols (SIAs), which include SNA salts, are
key components of PM$_{2.5}$(Li et al., 2016). According to Shang et al. (2020), when Beijing experienced severe winter
pollution episodes in 2013 and 2018, SNA accounted for 41% and 57% of PM$_{2.5}$ in those respective years. Here, we
investigate the role of atmospheric NH$_3$ in the formation of SIAs in Beijing by analyzing the relationship between NH$_3$ and
SNA concentrations during the observation period.

According to the study of Wei et al. (2023) conducted between 2013 and 2020, the SNA concentrations in Beijing exhibited
a significant downward trend. However, the proportion of SNA in PM$_{2.5}$ (mass concentration) did not change substantially




during this period. Table S2 lists the proportions of various SNA components in $PM_{2.5}$ (mass concentration) recorded in urban areas of Beijing for the years 2009, 2016, 2018, and 2019. In the summer and autumn of 2009, $SO_4^{2-}$ accounted for more than 50% of SNA content, considerably exceeding the concentrations of $NO_3^-$ and $NH_4^+$. However, by 2016, except for the summer season when $SO_4^{2-}$ was still the predominant component, $NO_3^-$ became the dominant component of the SNA

mass concentration. Over time, the proportion of $NH_4^+$ in the SNA mass concentration increased across multiple seasons. This finding indicates the necessity of controlling $NH_3$ and $NO_x$ concentrations to mitigate future $PM_{2.5}$ pollution.

Figure 4 depicts the relationship ion $NH_4^+$ in fine particulates and atmospheric $NH_3$. Overall, a positive correlation was identified between $NH_4^+$ and $NH_3$, indicating that variations in the concentration of the precursor gas $NH_3$ influenced the formation of $NH_4^+$. An increase in the $NH_3$ concentration led to a higher concentration of $NH_4^+$ in fine particulate matter, and

this effect was most pronounced in winter, in which the correlation between $NH_4^+$ and $NH_3$ was the strongest ($R^2 = 0.68$, $p <$ 0.01), and the average molar concentration ratio of $NH_4^+$ to $NH_3$ the highest. The seasonal differences in the response of aerosol $NH_4^+$ on atmospheric $NH_3$ may mainly be caused by variations in meteorological conditions, in addition to those in precursor gases of SNA For example, low temperature and high humidity promote the conversion of gaseous $NH_3$ to particulate $NH_4^+$(Wang et al., 2015). Thus, winter meteorological conditions may increase the formation of $NH_4^+$ in fine

particulate matter, which in turn exacerbated fine particulate pollution and haze formation.

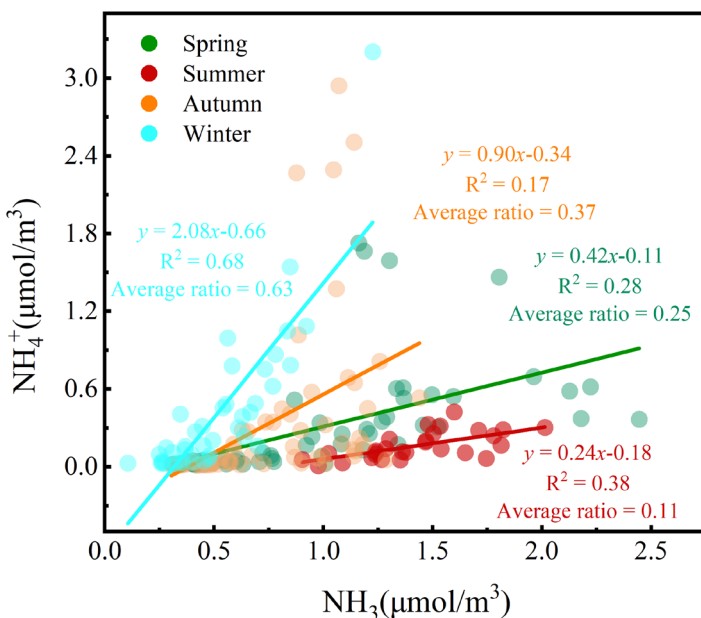

**Figure 4: Correlation between gaseous $NH_3$ and fine particulate ion $NH_4^+$.**



To gain further insights, the ISORROPIA II thermodynamic equilibrium model was employed to simulate and analyze the sensitivity of SNA in $PM_{2.5}$ to changes in precursor concentrations in each season. Concentrations of $SO_4^{2-}$ + $H_2SO_4$ (TS), $HNO_3$ + $NO_3^-$ (TN), and $NH_3$ + $NH_4^+$ (TA) were increased or reduced by up to 20%, and the changes in simulated concentrations relative to the baseline (without perturbation) were calculated. The simulation results revealed a strong correlation between the simulated and observed $NH_3$ concentrations ($R_{spring}$ = 0.89, $R_{summer}$ = 1.00, $R_{autumn}$ = 0.97, $R_{winter}$ = 0.98, p < 0.01), indicating the reliability of above observation-based results. The simulation results (Table S3) indicated that $NH_4^+$ was the least sensitive component to changes in the TA concentration in spring, autumn, and winter, suggesting that atmospheric $NH_3$ was not the limiting reactant for the generation of $(NH_4)_2SO_4$ and $NH_4NO_3$ in these seasons during the observation period. The responses of $NH_4^+$ to changes in TA and TN concentrations were less apparent in summer because $NH_4^+$ was mainly bound to $SO_4^{2-}$ rather than to $NO_3^-$ during this hot season, in which the high temperature was unfavorable for the generation and retention of $NH_4NO_3$. Furthermore, the increase in TS had an over or nearly (winter) proportional perturbation effect on $NH_4^+$, indicating the presence of sufficient $NH_3$ in the atmosphere of Beijing throughout the year. As also suggested by Su et al. (2021), the acidic components in the atmosphere in Beijing were sufficiently neutralized. Therefore, a relatively smaller reduction (say 20%) in $NH_3$ abundance seems not to be able to significantly lower the SNA levels in Beijing.

When $NH_3$ was abundant, the sensitivity of SNA content to changes in TA within the ±20% range was low (within ±2%, Table S3). Under such conditions, changes in TS and TN concentrations had much larger perturbation effects on SNA concentrations. However, the winter SNA concentrations mostly responded linearly to changes in TS and TN concentrations but nonlinearly to changes in the TA ones (Figure 5). If the TA concentrations were reduced by 60%, the rate of decrease in SNA content would have accelerated considerably, which were more pronounced in 2019. By contrast, the effect of changes in the TS concentration decreased after its reduction of over 40%. In a previous study based on nationwide measurements and simulation, Meng et al. (2022) suggested that SNA content in China can be more effectively controlled by reducing the concentrations of acidic gases ($SO_2$ and $NO_x$) in the atmosphere than by reducing the concentration of $NH_3$ by the same percentage. Additionally, Zheng et al. (2022) discovered that the joint control of $SO_2$ and $NO_x$ emissions is still the preferred method for reducing SNA concentrations in Central China, unless when acidic gas emissions are well controlled and the environmental chemical balance tends to favor the effective control of $NH_3$. Therefore, under the current atmospheric conditions, controlling acidic gas emissions is still a priority for reducing the $PM_{2.5}$ concentration in Beijing. Nevertheless, the cost of emissions reduction also increases with the progress of controlling $SO_2$ and $NO_x$ emissions. Although they provide similar abatement benefits, the cost of reducing $NH_3$ is only 10% of that required to reduce $NO_x$ (Gu et al., 2021). Thus, reducing $NH_3$ emissions should be prioritized as a means of improving future air quality in China.



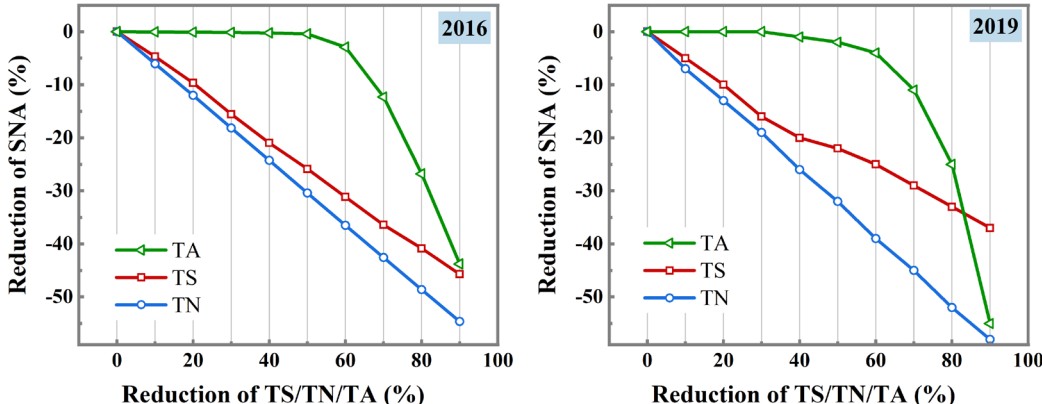

Figure 5: ISORROPIA predictions of percentage reduction in SNA mass concentration based on winter observations (charts display percentage reduction in SNA plotted against percentage reduction in TS, TN, and TA concentrations).

## 3 Conclusions

Over these 11 years, the $NH_3$ concentration in urban Beijing initially increased but subsequently decreased, resulting in a slight overall decrease. In particular, the annual average concentration of $NH_3$ in 2020 was 24% lower than that in 2009. The trend for $NH_3$ mixing ratios did not align with those for annual $NH_3$ emissions and the increasing trend indicated by satellite-based $NH_3$ monitoring data. These discrepancies highlight the complexity of $NH_3$ sources and removal processes in urban areas, which implies further challenges of performing atmospheric $NH_3$–related modeling and implementing future emission reduction strategies.

The long-term trend in $NH_3$ in urban Beijing was not significantly influenced by meteorological factors such as temperature. However, the seasonal variations in $NH_3$ mixing ratios were strongly influenced by temperature, with higher temperatures corresponding to higher $NH_3$ mixing ratios during warmer seasons. Regarding daily variations, $NH_3$ mixing ratios exhibited both positive and negative correlations with temperature but consistently exhibited a positive correlation with absolute humidity on most days. Across the observation years, the daily variations in $NH_3$ concentrations did not exhibit a consistent pattern across different seasons. In some cases, the patterns were even entirely opposite. By contrast, various meteorological factors and the daily variation patterns of other common air pollutants were mostly consistent across different years and seasons. Consequently, the factors influencing atmospheric $NH_3$ concentrations appeared to be more complex compared with those influencing other common air pollutants.

The concentrations of various $PM_{2.5}$ ion components in Beijing for the years 2009, 2016, 2018, and 2019 indicate that, apart from the summer season, the SNA content in Beijing was mainly dominated by sulfate (nearly 50%). Furthermore, the

proportion of ammonium in SNA content increased over time. An analysis of the neutralization levels of major acidic gases and a modeling analysis of perturbation indicated that an excessive concentration of $NH_3$ was maintained throughout the year in Beijing. The findings of the present study suggest that even though the concentrations of $SO_2$ and $NO_x$ in Beijing have decreased substantially over the past 2 decades, the current reduction of SIA remains less significant in response to $NH_3$ than

acid gases. Therefore, reducing acidic gas emissions is still a primary focus for controlling fine particulate matter pollution in the atmosphere. And in the future, more attention will be needed to focus on controlling $NH_3$ concentrations. Given that the trends in urban atmospheric $NH_3$ concentrations do not align with emissions trends, clarifying the relationship between them and identifying the sources of $NH_3$ in Beijing will play a crucial role in effectively reducing atmospheric $NH_3$ concentrations in the city.

In the present study, atmospheric $NH_3$ concentrations in urban Beijing were continuously monitored over a long period with high temporal resolution. However, it should be noted that the potential limitations of surface monitoring in representing urban or regional trends due to the uneven distribution of atmospheric $NH_3$ sources and the lack of vertical information. Similarly, with monitoring at a single site, it is necessary to verify whether the response measures are broadly applicable across the entire Beijing urban area. This will require further observational research on atmospheric $NH_3$ in urban Beijing in

the future. In addition, existing studies have demonstrated that emission inventories have underestimated atmospheric $NH_3$ emissions in the Beijing urban area (Xu et al., 2023), and the assessment results have varied across emission inventories (Chen et al., 2023). Additionally, given the limited research years of current emission inventories, the observed differences between long-term trends in monitored $NH_3$ concentrations and $NH_3$ emissions require continued attention in the future.

**Data availability**

For access to datasets, please contact Weili Lin.

**Author contributions**

Z. L., W.L., and X.X. designed the research, interpreted the data, and wrote the manuscript, Z.L., W.L., X.Z., Z.M. conducted the NH3 measurements and J.J., Y.Z., L.W. contributed the ion component data in PM2.5. The manuscript was written through the contributions of all authors. All authors have given approval to the final version of the manuscript.

**Competing interests**

The contact author has declared that none of the authors has any competing interests.



## Acknowledgments

The authors are grateful for the assistance of colleagues for sample collection.

## Financial support

This work was supported by the National Natural Science Foundation of China (grant no. 91744206, grant no. 42175128), the Beijing Municipal Science and Technology Commission (grant no. Z181100005418016), and the CAMS technology and development fund project(2022KJ005).

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
