# Peer review of "Atmospheric NH3 in urban Beijing: long-term variations and implications for secondary inorganic aerosol control"

_EGUsphere, 2024_

## Author Comment (AC1)

**Response to comments on "Atmospheric NH₃ in urban Beijing: long-term variations and implications for secondary inorganic aerosol control"**

Ziru Lan[1], Xiaoyi Zhang[2], Weili Lin[1], Xiaobin Xu[2], Zhiqiang Ma[3], Jun Jin[1], Lingyan Wu[2], Yangmei Zhang[2]

[1]Key Laboratory of Ecology and Environment in Minority Areas, Minzu University of China, National Ethnic Affairs Commission, Beijing 100081, China

[2]Institute of Atmospheric Composition, Chinese Academy of Meteorological Science, Beijing 100081, China

[3]Institute of Urban Meteorology, China Meteorological Administration, Beijing 100089, China

*Correspondence to*: Weili Lin(linwl@muc.edu.cn)

General comments:

This paper describes the implications of standard/state-of-the-art NH₃ measurements taken in Beijing between 2009 and 2020. NH₃ is an air pollutant that is principally emitted from human activities, affecting ecosystems, air quality and climate. The manuscript is a good fit for the ACP audience. I found the methods used to be generally sound relative to the conclusions drawn, and the paper is well written and clear. I would recommend this article for publication after the following comments are addressed (some major, some minor).

----- We greatly appreciate your time spent reviewing our manuscript and providing constructive comments. We have revised the manuscript according to your suggestions.

Specific comments:

Major:

I feel that there should be better acknowledgement of the uncertainty inherent in relocating the measurement in 2017. While there have been references showing that the NH3 vertical profile is relatively constant up to 300 m in Beijing, a measurement on the 14 story could be on the cusp of surpassing 300 m. There has also been recorded greater variability in the vertical profile in other locations. The manuscript raises that there is spatial variability in NH3, so it's not clear why this couldn't also extend to measurements of the vertical profile (in other words, could the vertical profile of NH3 differ in

different parts of Beijing?). In my opinion, it would be better to acknowledge the uncertainty associated with this move, raise plausible hypotheses, and indicate that there is too much uncertainty to draw confident conclusions about the drop in 2017. Could emissions or changes in the urban topography have contributed? These aren't discussed here but might also be relevant. The authors may also consider splitting the dataset in 2017, but it is probably worth maintaining the dataset as continuous because the horizontal distance between the two locations is so small.

----- Thank you for your suggestion. We acknowledge the necessity of addressing the uncertainties associated with our observations and have made the following modifications accordingly: ①We have added detailed descriptions regarding the heights of the observation locations: The ground-floor elevations of both buildings are 56 m, and the observation heights above the ground are 10 m on the 3rd floor and 56 m on the 14th floor. Studies of vertical observations of atmospheric $NH_3$ in Beijing have been conducted at the Beijing Meteorological Tower, which is situated in an urban area. Both locations are approximately 6.7 km apart and have the same elevation of 56 m. Given that the 14th floor of our observation site is 56 m above the ground, which is considerably lower than 300 m, we believe that these vertical profile results are applicable as a reference for our study. ②We have added a discussion on splitting the dataset into three segments, comparing the segmented data results with those from the continuous dataset. Overall, the revised Section 3.1 now reads as follows:

[revised manuscript text omitted]

I found it strange that there is an anticorrelation between NH3 and temperature (on an average daily basis) and feel that the authors should include more interpretation of this finding. Are there any other papers showing an NH3-T anticorrelation?

----- Several studies have examined the diurnal variations of $NH_3$, showing similar patterns during spring, summer, and autumn as observed in our study, but without discussing their relationship with temperature (Buijsman et al., 1998; Sharma et al., 2014; Gu et al., 2022). In our previous research, the urban site exhibited a negative correlation between $NH_3$ concentrations and temperature's diurnal characteristics during summer, autumn, and winter, whereas the rural site showed a positive correlation (Lan et al., 2021). Buijsman et al. (1998) suggested that lower daytime $NH_3$ concentrations in high emission areas might be due to higher wind speeds and more favorable mixing conditions, with $NH_3$ accumulating during the night within a shallower boundary layer. Our current manuscript delves deeper into these issues, with the revised paragraph as follows:

Several studies have suggested that temperature plays a pivotal role in driving diurnal variations in atmospheric $NH_3$ concentrations (Clarisse et al., 2021; Langford et al., 1992). However, the present study shows that $NH_3$ concentrations are significantly influenced by temperature across seasonal changes (Figure 3); in terms of diurnal patterns, the days showing positive and negative correlations between $NH_3$ concentrations and temperature each constitute nearly half of the effective observation days (Figure S11). The mean diurnal variations in different seasons typically exhibit lower concentrations during the day and higher at night in spring, summer, and autumn (Figure S12). This difference suggests that correlations observed on a seasonal (climatic scale) tend to obscure lower-frequency data relationships, with daily variations in $NH_3$ concentrations being more influenced by the transition from day to night (meteorological or weather scale). This highlights the complex factors affecting $NH_3$ concentrations in the urban areas of Beijing and underscores the importance of high temporal resolution in observations. Several studies have noted a reduction in $NH_3$ concentrations during daylight hours (Buijsman et al., 1998; Sharma et al., 2014; Gu et al., 2022; Lan et al., 2021). Increased temperatures during the day promote the volatilization of $NH_3$, but lower daytime concentrations may result from higher wind speeds and more favorable mixing conditions, whereas at night, $NH_3$ tends to accumulate within a shallower boundary layer (Buijsman et al., 1998). The diurnal variation of the boundary layer height in Beijing exhibits a single-peak pattern, rising rapidly from 6:00 to 8:00, reaching its peak between 14:00 and 15:00, then declining sharply, and stabilizing after 18:00 to 20:00 (Figure S12). During the daytime, $NH_3$ concentrations are influenced by a combination of temperature (which promotes emissions) and changes in the boundary layer height (which causes dilution), with the valley value of $NH_3$ concentrations lagging

behind the peak times of boundary layer height and temperature. Moreover, during spring, summer, and autumn, the continuous decline in $NH_3$ concentrations during the daytime indicates that vertical mixing transport has a limited impact on atmospheric $NH_3$ in the urban areas of Beijing. In winter, the evaporation of dew or frost in the early morning leads to a rapid increase in $H_2O$ and $NH_3$ concentrations (Wentworth et al., 2016), while the effects of afternoon temperature and vertical mixing dilution are comparable, keeping $NH_3$ concentrations relatively stable.

[Figure]

Figure S12. Average diurnal variations in $NH_3$, temperature, absolute humidity and boundary layer height in different seasons in Beijing urban area. Boundary layer height data are from the ERA5 global atmospheric reanalysis (Hersbach et al., 2023)

Figure 3 - To my eye there generally appears to be a positive correlation between air T and NH3. The text starting on L241 also suggests there is a pretty even balance of days with positive or negative correlation between NH3 and air T (my understanding is that this is correlating hours within a day). It seems to me that there could plausibly be a weak negative correlation on a daily average basis, but I don't understand how there are strong negative correlations between air T and NH3 for most seasons in L246. Is this correct? Otherwise, could the interpretation among these relationships between air T and NH3 be expanded to clarify?

----- It is widely accepted that $NH_3$ concentration positively correlates with temperature, and our study's

findings on the seasonal variations of NH$_3$ concentrations also support this consensus. During the observation period, the temperature in Beijing followed the seasonal sequence of summer (being the warmest), spring, autumn, and winter. The rankings of NH$_3$ concentrations across the seasons were consistent with this trend. In Figure 3, we calculated average values of NH$_3$ concentrations and temperatures in each season annually. We found a significant correlation only in winter (R = 0.38, p < 0.05), with no significant correlations in other seasons. Line 241 discusses the results from Figure S11, where we analyzed the hourly averages of NH$_3$ concentration and their correlation with temperature on diurnal variations. Line 246 presents the correlation of average daily variations between NH$_3$ concentrations and temperature over an eleven-year observation period (Figure S12). The strong negative correlation reported on L246 might seem counterintuitive. However, such diurnal variation patterns are more prevalent during the seasons with higher concentrations (summer, spring, and autumn), and tend to obscure individual daily patterns after averaging. This seemingly contradictory result reveals the complexity and difference between climatic scale and weather scale factors. We have added the following content in the original text for further explanation: However, the present study shows that NH$_3$ concentrations are significantly influenced by temperature across seasonal changes (Figure 3); in terms of diurnal patterns, the days showing positive and negative correlations between NH$_3$ concentrations and temperature each constitute nearly half of the effective observation days (Figure S11). The mean diurnal variations in different seasons typically exhibit lower concentrations during the day and higher at night in spring, summer, and autumn (Figure S12). This difference suggests that correlations observed on a seasonal (climatic scale) tend to obscure lower-frequency data relationships, with daily variations in NH$_3$ concentrations being more influenced by the transition from day to night (meteorological or weather scale). This highlights the complex factors affecting NH$_3$ concentrations in the urban areas of Beijing and underscores the importance of high temporal resolution in observations.

Minor:

Generally, I found the abstract to be a well written and succinct summary that draws out keen points of interest for this manuscript. My only suggestion is to strengthen the motivation for controlling NH3 concentrations in the last sentence. The preceding sentence expresses that SIA concentrations are not very sensitive to NH3, so it could be helpful to provide additional rationale (for example, that reducing

NH3 is cheaper? It's more effective once NH3 has been reduced to some extent?—both discussed in the paper).

----- Thank you for your suggestion. We have revised the last sentence of the abstract as follows: "Although reducing $NH_3$ concentrations can improve air quality during winter, controlling acid gas concentrations has a greater effect than controlling $NH_3$ concentrations on reducing SIA concentrations, until $NH_3$ and acidic gas concentrations are reduced below 80% of their current levels. Nevertheless, the increase in the proportion of ammonium salts in SIAs during the observation period indicates that future control measures for $NH_3$ concentrations may need to be prioritized in Beijing."

L38 – suggest clarifying that "particulate matter 2.5" is "particulate matter with a diameter less than 2.5 μm in size".

----- Done.

Sentence starting on L36 ("However, long-term…") – I think that this paragraph motivates the utility of long-term trends in ground-based atmospheric NH3 well, but it would benefit from expanding on the urban aspect of this study. For example, are NNMDN sites generally in rural areas?

----- We have adjusted this section to make it more relevant to the content of the study: In China, according to the monitoring results from the Nationwide Nitrogen Deposition Monitoring Network (NNDMN), $NH_3$ concentrations at 12 urban sites and 43 rural sites increased by approximately 80% from 2011 to 201. Satellite data analysis by Dong et al. (2023) indicated a significant increase (~32%) in $NH_3$ vertical column densities in China from 2008 to 2019. In the North China Plain, a hotspot for global $NH_3$ emissions, Luo et al. (2020) found a rapid increase in urban $NH_3$ concentration from 2011 to 2018. Wen et al. (2024) found a 26% decrease in Beijing $NH_3$ concentrations from August 2005 to August 2020, and a 50% increase from January 2005 to January 2020. Currently, long-term ground-based observations of atmospheric $NH_3$ at high temporal resolution are relatively rare in China, and the contrasting trends between $NH_3$ emissions, satellite and in-situ measured concentrations in urban areas have not been fully explore.

L78 – Describe source of emission data in Figure S1 either in the main text or figure caption.

----- We have added a description of the sources of the emission data: Figure S1. $NH_3$ emission in and around Beijing (a) and topographic map of Beijing (b). The $NH_3$ emissions data represent the total for 2017, sourced from emission inventories (Huang et al., 2012; Kang et al., 2016).

L79 – I assume that the inlet is outside of the building—could you clarify?

----- The inlet is located outside the building. In our setup, the inlet lines are positioned approximately 1.5 m above the floor level of the building. It is installed through a hole in the window using a polytetrafluoroethylene (PTFE) tube that connects to the instrument, with the hole fitting snugly around the sampling tube. The conduit extends outward by about 15 cm, and to prevent water ingress, there is a downward-sloping shield above the conduit. To clarify further, I have added details about the sampling setup in the second paragraph of Section 2.1: At CMA site, the air had been drained into an air-conditioned room with a 4.5 m long Teflon line and the inlet height is 1.8 m above the rooftop (about 12 m above ground level). At MUC site, air is introduced from outside the sealed window through a borehole, with the air inlet extending 20-30 cm outside the window. Since it is on the 14th floor, the air outside the building flows smoothly.

L80 – Later in the manuscript it is noted that there is not a strong vertical profile of NH3 in Beijing (although this has been observed in other locations). It may be worth mentioning that here as well.

----- Thank you for your suggestion. Presenting the vertical profile characteristics of $NH_3$ in Beijing here would be beneficial for understanding the reliability of our data. However, to maintain the focus and flow of this section, we have opted to discuss this in the data analysis section later in the manuscript.

L100 – Do you have any thoughts on why the slope is higher in this study than in Zhang et al. 2021?

----- The discrepancies in the observation results can be attributed to differences in the timing of

instrument comparisons between Zhang et al. and our study, as well as variations in instrument conditions over time. Furthermore, the longer duration of observations in our study exposes our instruments to a broader range of environmental conditions, which may also contribute to the observed differences in slopes.

L112 – Could you please clarify why you switched the source of the met data?

----- Due to the unavailability of meteorological data from the Haidian weather station, which is closest to our observation site before 2012, we utilized meteorological data from the Capital Airport for continuity in data analysis. By comparing the correlation of meteorological data(Figure S5), we believe that it is feasible to use temperature and relative humidity data from the Capital Airport to supplement the analysis in this study.

L122 – Could you please clarify where is the Chinese Academy of Meteorological Sciences relative to the other measurement locations of interest?

----- We apologize for the ambiguity in our expression that may have caused a misunderstanding. The original sentence: "$PM_{2.5}$ samples collected on filters were analyzed for ion components ($Na^+$, $SO_4^{2-}$, $NH_4^+$, $NO_3^-$, $Cl^-$, $Ca^{2+}$, $K^+$, and $Mg^{2+}$) at the Chinese Academy of Meteorological Sciences, resulting in the acquisition of 184 data sets." Here, the Chinese Academy of Meteorological Sciences was the location where the samples were analyzed, not where the monitoring occurred. We have revised this sentence to: "The collected $PM_{2.5}$ samples on filters were subsequently sent to the Chinese Academy of Meteorological Sciences for chemical analysis of ion components ($Na^+$, $SO_4^{2-}$, $NH_4^+$, $NO_3^-$, $Cl^-$, $Ca^{2+}$, $K^+$, and $Mg^{2+}$), from which 184 data sets were obtained."

Section 2.2.1 – It would be helpful to briefly state why this particular method is being used (to gap fill?).

----- Thank you for your comment. The use of EEMD is not intended for filling data gaps, but rather for a clearer analysis of long-term trends. Traditional trend analysis tools have limitations, especially when dealing with weak and non-linear trends. For instance, the linear regression method does not represent

the true trends of non-linear data accurately. The combined use of the Mann-Kendall test and Theil-Sen (TS) methods for long-term trend analysis is affected by seasonal fluctuations in the data series. In contrast, Ensemble Empirical Mode Decomposition (EEMD) is an advanced time-frequency analysis technique that is well-suited for extracting trends from time series that are non-linear and exhibit irregular cycles. The temporal characteristics of atmospheric NH3 are non-linear and non-stationary, hence using EEMD allows for a more comprehensive understanding of underlying trends and cycles often masked in traditional analysis methods. We have included additional descriptions of EEMD in our manuscript: Compared to traditional long-term trend analysis tools, EEMD shows greater stability in decomposing non-linear and non-stationary data series. It is unaffected by the seasonal variations in data series and can resolve non-linear changing trends, making it more suitable for environmental data trend analysis. Therefore, using EEMD enables more accurate extraction of genuine signal variations.

Section 2.2.2 – Could you please describe the model configuration? For example, are the PM5 ion measurements used to run the model? It might be helpful to make that connection and remind the reader of that available data.

----- We reorganized this section: The ISORROPIA-II model is mainly used to simulate the physical state and concentration of inorganic components of the aerosol system at thermodynamic equilibrium. A distinct advantage of the ISORROPIA-II model over other thermodynamic models is the inclusion of the $K^+$, $Ca^{2+}$, and $Mg^{2+}$ ions in the calculations, and taking these components into account significantly improves the accuracy of the model simulations (Allen et al., 2015). Additionally, the high precision and computational efficiency of the ISORROPIA II mode have been widely demonstrated (Fountoukis and Nenes, 2007). To assess the sensitivity of sulfate, nitrate, and ammonium (SNA) to changes in precursor concentrations, the present study employed the ISORROPIA II thermodynamic equilibrium, version 2.3 (http://isorropia.epfl.ch). The model was run in "forward + metastable" mode, taking inputs such as temperature (unit is k), relative humidity (up to 1), and concentrations of particulate components ($SO_4^{2-}$、 $Cl^- + HCl$、$NO_3^- + HNO_3$、$NH_4^+ + NH_3$、$Na^+$、$K^+$、$Ca^{2+}$ and $Mg^{2+}$) expressed in μg m$^{-3}$ for calculations.

Table S1 – It's not totally clear what's gained from comparing this study's results to other studies located

in different parts of the world. I am also unclear on what the numbers in the "This study" column represent – are those the averages from the same timeframe as the study that they are being compared with? Please expand/clarify.

----- Thank you for your comment. This column was intended to compare $NH_3$ concentrations during the same period with other studies. However, we agree that this column does not significantly contribute to the analysis, so we have decided to remove it.

L227 – I don't understand how the variation is being calculated. It is described as "average annual temperature" but then specified by season. Please clarify?

----- Thank you for your comment highlighting the ambiguity in our expression. In our manuscript, the term "average annual temperature" refers to the calculation of average temperatures for each season within a year. We calculate the average temperature for each season and then discuss the variation coefficients of these seasonal averages for each year. Accordingly, we have revised the original text from "The interannual trends pertaining to temperature and $NH_3$ mixing ratios across multiple seasons (Figure 3) revealed that temperature remained stable in summer and autumn over the years; when calculated in Kelvin, the average annual temperature exhibited variation coefficients of 0.42% in spring, 0.15% in summer, 0.17% in autumn, and 0.51% in winter." to "The interannual trends for temperature and $NH_3$ mixing ratios across multiple seasons (Figure 3) revealed that temperature remained stable in summer and autumn over the years; when calculated in Kelvin, the average seasonal temperatures exhibited interannual variation coefficients of 0.42% in spring, 0.15% in summer, 0.17% in autumn, and 0.51% in winter."

Generally, I felt that the authors could be clearer about the specificity of their results for Beijing (as appropriate), in particular in the conclusions and when discussing the policy prioritization in the abstract (e.g. "measures to control NH3 concentrations should be prioritized [in Beijing].").

----- Thank you for your suggestion; we have made the necessary modifications.

L367 – reiterate that this is for Beijing——"Therefore, reducing acidic gas emissions is still a primary focus for controlling fine particulate matter pollution [in Beijing].

-----Done.

L376 – Could you recall the evidence supporting "And in the future, more attention will be needed to focus on controlling NH3" ?

----- Since this sentence is speculative and seems inappropriate for the conclusion section, we have decided to remove it.

Technical comments

There is often a missing space between a word and the opening parenthesis for the citations

----- We were really sorry for our careless mistakes. Thank you for your reminder.

Figure S3 caption – It's a little confusing that the subpart labels first follow and then precede the description of the content. It would be more intuitive to keep it consistent (e.g. "Monthly (a) and annual (b) variations and correlations between satellite observations… 116.5E) (c) and the average observations… ~118.5E) (d)."

----- Thank you for your suggestions. We have revised the text as follows: Monthly (a) and annual (b) variations and correlations between satellite observations during the observation period at the grid points around the monitoring stations (39.5°N, 116.5°E and 40.5°N, 116.5°E) (c) and the average observations in the region selected for the present study (36.5°N~42.5°N, 113.5°E~118.5°E) (d).

Consider referencing Figure 1 earlier, where the results are first mentioned (I believe in L172?)

----- Done.

L374 I found this wording awkward: "the current reduction of SIA remains less significant in response to NH3 than acid gases." Consider: "SIA formation is more sensitive to acid gases than NH3.

----- This has been done as suggested.

---

## Author Comment (AC2)

**Response to comments on "Atmospheric NH₃ in urban Beijing: long-term variations and implications for secondary inorganic aerosol control"**

Ziru Lan[1], Xiaoyi Zhang[2], Weili Lin[1], Xiaobin Xu[2], Zhiqiang Ma[3], Jun Jin[1], Lingyan Wu[2], Yangmei Zhang[2]

[1]Key Laboratory of Ecology and Environment in Minority Areas, Minzu University of China, National Ethnic Affairs Commission, Beijing 100081, China

[2]Institute of Atmospheric Composition, Chinese Academy of Meteorological Science, Beijing 100081, China

[3]Institute of Urban Meteorology, China Meteorological Administration, Beijing 100089, China

*Correspondence to*: Weili Lin(linwl@muc.edu.cn)

This study presents 11-year NH3 data in an urban environment and explored the long-term trends of NH3, the influence meteorological variables played on NH3, and the role NH3 played on secondary inorganic aerosol formation. While the data presented here are useful, the analysis can be improved. More importantly, the discuss and presenting quality needs significant improvement. More specific comments are provided below.

---- We are very grateful for your critical comments and thoughtful suggestions. Based on your comments, we have made detailed revisions to the manuscript.

Abstract:

Abstract needs a significant revision to better summarize major findings. Too many general statements, but lack of specific results.

----- We have revised the abstract, and the specific modifications are as follows:

Line 13, better replace "The total average" with "The 11-year average".

----- Thank you for your suggestion. We have updated the original sentence to "The 11-year average NH₃ mixing ratio was 26.9 ± 19.3 ppb".

Lines 14-15: Why not show the total percentage increase between 2009-2017 and percentage decrease between 2017-2020? This way can better reflect the two contrasting trends during the 11-year period.

----- We added quantitative descriptions to the sentence. The original statement has been amended to "NH$_3$ mixing ratios initially increased and peaked in 2017 but subsequently decreased. From 2009 to 2017, NH$_3$ mixing ratios increased by 50%, while there was a decrease of 49% in 2020 compared to 2017."

Line 17: This sentence does not provide any useful information. Most pollutants would have seasonal and diurnal variations. You need to specify what kind of seasonal and diurnal patterns.

----- We agree that the sentence was somewhat awkward, and have removed it in the revised version.

Liens 18-19: the non-linear relation is well known in literature. You need to show some specific results. Same comment for several other sentences in this section.

----- We have added specific results to the sentences: Thermodynamic modeling revealed the nonlinear response of SIAs to NH$_3$, with increased sensitivity to NH$_3$ when its concentration decreases by 60%. Although reducing NH$_3$ concentrations can improve air quality during winter, controlling acid gas concentrations has a greater effect than controlling NH$_3$ concentrations on reducing SIA concentrations, until NH$_3$ and acidic gas concentrations are reduced below 80% of their current levels. Nevertheless, the increase in the proportion of ammonium salts in SIAs during the observation period indicates that future control measures for NH$_3$ concentrations may need to be prioritized in Beijing.

Introduction

This section also needs some major rewriting. The topic of this research is on urban NH3 and its long-term trends. Thus, the Induction should cover these areas: (i) brief discussion on the important role NH3 played in various research areas (this is covered in the current Introduction, but could be simplified since such materials are rich in literature); (2) brief discussion on the major sources of NH3 in urban environments and major debate on this topic in literature, noting that existing studies have different opinion on the major sources (this is not mentioned at all in the current Introduction and should be added in the revision. Such materials may help explain the trends in Section 3.1); and (3) brief discussion on the studies of NH3 long-term trends, first worldwide (see Yao and Zhang. 2009, ACS Omega, 4, 22133-22142, as an example), then China, then Beijing. Then point out the knowledge gaps based on the

summary of the knowledge presented above. Finally present the goals of this study.

----- Thank you for your suggestions. We have simplified the discussion on the environmental impact of $NH_3$, expanded the discussion on urban $NH_3$ sources, and re-summarized the studies on long-term $NH_3$ trends. Below is the revised introduction section:

Excessive input of anthropogenic nitrogen into the environment can directly harm ecosystems and influence climate change. As the most abundant alkaline trace gas in the atmosphere, $NH_3$ interacts with the oxidized products of atmospheric acidic gases to form secondary aerosols, which considerably affect the radiative balance of the atmosphere and ai. Over the years, China has been committed to controlling air pollution and has effectively managed the emissions of primary pollutants such as sulfur dioxide ($SO_2$) and nitrogen oxides ($NO_x$). However, particulate matter 2.5 ($PM_{2.5}$, particulate matter with a diameter less than 2.5 μm in size) pollution is still a severe problem. Research on controlling $SO_2$ and $NO_x$ emissions indicate that controlling $NH_3$ emissions is the most economically effective way for reducing $PM_{2.5}$. However, the effectiveness of $NH_3$ reduction varies by region, and there is still debate regarding the efficacy of $NH_3$ reduction measures.

Anthropogenic sources are the primary contributors to atmospheric $NH_3$ emissions. In China, agricultural sources dominate, accounting for approximately 80% of total emissions. However, the contribution of non-agricultural sources in urban areas is considered significant. Studies indicate that over 30% of $NH_3$ emissions observed in urban areas can be attributed to traffic. Nevertheless, some research suggests that biogenic sources (primarily green spaces) predominate in urban and account for approximately 60% of emissions, while the contribution from traffic sources is negligible. The complexity of urban $NH_3$ sources results in intricate variability in its atmospheric characteristics.

Long-term observations are important for analyzing the environmental impacts and control strategies of atmospheric $NH_3$. In Europe, North America and Asia, countries have conducted studies on $NH_3$ variations over a period of 5 years or more. In most of these regions, $NH_3$ concentrations have either remained stable or have exhibited an increasing trend. Satellite observations detected rising global atmospheric $NH_3$ concentrations, influenced by reductions in acidic gas emissions, temperature increases, and the rising use of chemical fertilizers. In China, according to the monitoring results from the Nationwide Nitrogen Deposition Monitoring Network (NNDMN), $NH_3$ concentrations at 12 urban sites

and 43 rural sites increased by approximately 80% from 2011 to 2018. Satellite data analysis by Dong et al. (2023) indicated a significant increase (~32%) in $NH_3$ vertical column densities in China from 2008 to 2019. In the North China Plain, a hotspot for global $NH_3$ emissions, Luo et al. (2020) found a rapid increase in urban $NH_3$ concentrations from 2011 to 2018. Wen et al. (2024) found a 26% decrease in Beijing $NH_3$ concentrations from August 2005 to August 2020, and a 50% increase from January 2005 to January 2024. Currently, long-term ground-based observations of atmospheric $NH_3$ at high temporal resolution are relatively rare in China, and the contrasting trends between $NH_3$ emissions, satellite and in-situ measured concentrations in urban areas have not been fully explored

The present study examined high temporal resolution $NH_3$ observations at the surface in urban Beijing from 2009 to 2020. Using data from emission inventories, satellite observations, meteorological elements, concentrations of various types of atmospheric pollutants, and particle ion composition, the present study aims to obtain the characteristics of long-term variations, influencing factors, and the contributions of $NH_3$ to particle formation in the atmosphere of Beijing. Analyzing long-term $NH_3$ observations can help to understand how changes in $NH_3$ concentrations have affected atmospheric pollution in the past. This knowledge is crucial for predicting future atmospheric pollution and formulating effective environmental policies. Additionally, it provides a scientific basis and reference for developing future $NH_3$ control strategies.

Materials and methods

Line 79-80: need to specify the height above the ground of the two measurement sites (after the third floor and 14th floor).

----- We added descriptions of the elevations: The ground-floor elevations of both buildings are 56 m, and the observation heights above the ground are 10 m on the 3rd floor and 56 m on the 14th floor.

Line 90: change "subjected" to "subject"

----- Done.

Line 126: change "2.1 Methods" to "2.2 Data analysis methods". You already used Methods for Section 2, here you need to use a more specific sub-section title.

----- This has been done as suggested.

Line 134-135: This statement is not accurate. EEMD has also been used in air-quality trend analysis studies in more recent years (for example: Yao and Zhang, 2016. ACP 16, 11465-11475.   Wang et al., 2022. Environment International, 159, 107031. Wang and Zhang, 2023, Environmental Pollution, 333, 122079). You need to cite more relevant studies instead of not-so-relevant studies.

----- We have rewritten this sentence according to your suggestion: Currently, EEMD has been used in studies on air-quality trend analysis (Yao and Zhang, 2016; Fu et al., 2020; Wang et al., 2022; Wang and Zhang, 2023).

3 Results and discussion

This section needs a better organization and more in-depth analysis.

A large portion of Section 3.1 is used to compare the NH3 data with literature, and the discussions on the trends and associated causes are limited. While comparing to literature data is needed, it should not be the main focus of the discussion. Besides, from Lines 190-191: if this statement is true, then it means that the uncertainties in the obtained trends (due to changing location and measurement height) are larger than the actual trends, making your discuss on trends meaningless. I would recommend reorganizing Section 3.1 in this order: First present the trends from the monitored data using a quantitative statement, e.g., either using annual decreasing/increasing rate, or percentage decrease/increase during a period. Split the 11-year period into two periods since contrasting trends were observed during the whole measurement period. Use quantitative statements wherever possible and show the significance level of the trends. Then present the trends generated from the satellite data using the same rules as described above. Then discuss the similarities and differences between these two sets of trends, only at this stage you need to cite literature data to support your results and/or provide explanations on the causes of the differences between different data sets.

----- Thank you for your constructive comments and recommendations, we have reorganized Section 3.1: From June 2009 to July 2020, the hourly average mixing ratio of atmospheric $NH_3$ in Beijing was 26.9 $\pm$ 19.3 ppb (median, 23.5 ppb). Table S1 summarizes results from various $NH_3$ monitoring studies

[revised manuscript text omitted]

In general, long-term trends are mainly caused by emission changes and to a much less degree by meteorological factors. After discussing the trends in Section 3.1, you can then discuss driving factors of these trends in section 3.2, first focus on emission and then on meteorology. Emission inventory related discussion in Section 3.1 can be moved to section 3.2 to support the discussion. See Lin et al. (2022 ACP, 22, 16073-16090) to get more ideas related to this comment.

----- We have revised the title of section 3.2 to "Influences on variation characteristics of NH₃" and have added a discussion on the impact of emissions at the beginning of this section. The specific additions are as follows:

NH$_3$ emissions directly affect the variations in atmospheric NH$_3$ concentrations. The emission inventory data obtained (Figure 1) indicate that from 2009 to 2014, the total NH$_3$ emissions in Beijing remained stable, peaking in 2012. After 2014, NH$_3$ emissions in Beijing rapidly decreased, declining by 25% from 2012 to 2017. However, during this period of declining emissions, the NH$_3$ mixing ratio in Beijing exhibited an increasing trend. Similar phenomena have been reported by studies conducted outside of China. For instance, in Scotland, NH$_3$ emissions decreased by approximately 15% from 1990 to 2003, whereas atmospheric NH$_3$ concentrations increased. In Hungary, NH$_3$ emissions were estimated to have decreased by 50% from 1983 to 1993; however, NH$_3$ concentrations exhibited a slight upward trend during this monitoring period. A possible reason for these differences between NH$_3$ emissions and concentrations could be the significant reduction in the concentrations of SO$_2$ and NO$_x$, which reduced the amount of atmospheric NH$_3$ neutralized by acid gases. Over the past 2 decades, Beijing has implemented a series of strict measures to control air pollution and has achieved considerable success. The concentrations of SO$_2$, NO$_2$, CO, PM$_{10}$, and PM$_{2.5}$ in Beijing all exhibited decreasing trends; in particular, the concentration of SO$_2$ decreased by 88% from 2009 to 2020 (Figure 2).

To discuss the influence of chemical loss on the annual increase in NH$_3$ concentrations, the present study referred to research by Yao et al. (2019), assuming that NH$_4^+$ is uniformly distributed in the urban area of Beijing and that changes in NH$_4^+$ concentrations directly affect atmospheric NH$_3$ concentrations on a 1:1 basis. By calculating the change in NH$_4^+$ concentration relative to the baseline year, we adjust the atmospheric NH$_3$ concentrations. The present study set 2009 as the baseline year, using the annual average NH$_4^+$ concentration observed by Cheng (2021) in the urban area of Beijing to calculate the adjusted NH$_3$ concentrations from 2009 to 2017. The calculations show (Figure S10) that overall, the original NH$_3$ concentration in 2017 was 50% higher than in 2009, and the adjusted NH$_3$ concentration was 46% higher. Therefore, changes in chemical losses have a limited impact on the increased trend of NH$_3$ concentrations, and the discrepancy between NH$_3$ concentrations and emission trends may be due to imperfections in the emission inventory.

[Figure]

Figure S10. Annual averages of atmospheric NH$_3$, NH$_4^+$ in PM$_{2.5}$, and adjusted atmospheric NH$_3$ and NH$_3$ emissions in Beijing urban area from 2010 to 2017

Section 3.3 also has too many introductory materials in the first two paragraphs. Follow this rule, in the Results section, present your own results first and use literature data to support your discussion, instead of summarizing literature results separately (which really belong to Introduction section).

----- Thank you for your suggestion. We have revised Section 3.3, removing the introductory material and reorganizing the first paragraph: The present study investigates the role of atmospheric NH$_3$ in the formation of SIAs in Beijing by analyzing the relationship between NH$_3$ and SNA concentrations during the observation period. According to the study of Wei et al. (2023) conducted between 2013 and 2020, the SNA concentrations in Beijing exhibited a significant downward trend. However, the proportion of SNA in PM$_{2.5}$ (mass concentration) did not change substantially during this period. Table S2 lists the proportions of various SNA components in PM$_{2.5}$ (mass concentration) recorded in urban areas of Beijing for the years 2009, 2016, 2018, and 2019. In the summer and autumn of 2009, SO$_4^{2-}$ accounted for more than 50% of SNA content, considerably exceeding the concentrations of NO$_3^-$ and NH$_4^+$. However, by 2016, except for the summer season when SO$_4^{2-}$ was still the predominant component, NO$_3^-$ became the dominant component of the SNA mass concentration. Over time, the proportion of NH$_4^+$ in the SNA mass concentration increased across multiple seasons. Wen et al. (2024) and Cheng (2021) have also observed this phenomenon in urban Beijing. These findings indicate the necessity of controlling NH$_3$ and NO$_x$ concentrations to mitigate future PM$_{2.5}$ pollution.

Conclusions

Line 353: "3" should be "4". Avoid such simple typos.

----- Thanks for your careful checks, we are sorry for our carelessness, the typo has been corrected.

Lines 354-356: Preneet the two different trends (in two periods) using a quantitative statement.

----- We have revised the original text to read: Over these 11 years, the $NH_3$ concentration in urban Beijing initially increased by 50% in the first 8 years but subsequently decreased by 49% in the following 3 years.

Line 356: have you tried to identify the actual causes of such discrepancies between NH3 concentration and NH3 emission?    See possible causes on the same topic in Yao and Zhang (2009, ACS Omega, 4, 22133-22142).

----- Thank you very much for your suggestions. We have added a discussion on potential causes in Section 3.2, the details of which are displayed in the comments above.

---

## Author Response (AR2)

**Response to comments on *Atmospheric NH₃ in urban Beijing: long-term variations and implications for secondary inorganic aerosol control***

Ziru Lan[1], Xiaoyi Zhang[2], Weili Lin[1], Xiaobin Xu[2], Zhiqiang Ma[3], Jun Jin[1], Lingyan Wu[2], Yangmei Zhang[2]

*[1] Key Laboratory of Ecology and Environment in Minority Areas, Minzu University of China, National Ethnic Affairs Commission, Beijing 100081, China*

*[2] Institute of Atmospheric Composition, Chinese Academy of Meteorological Science, Beijing 100081, China*

*[3] Institute of Urban Meteorology, China Meteorological Administration, Beijing 100089, China*

---We sincerely appreciate the time and effort invested by the referees and editor in evaluating our manuscript. We revised our manuscript according to the comments.

**Response to comments by Anonymous referee #2**

The revised manuscript has been significantly improved in scientific quality, although the presentation quality can still be improved. I only listed some grammar issues in Abstract and Introduction below, and I recommend the authors to carefully proofread the whole manuscript.

----- We greatly appreciate your time reviewing our manuscript, we have revised the original text and carefully proofread the entire manuscript according to your suggestions.

Line 12: change "their long-term behaviors" to "its long-term behaviors"

Lines 14-16: combine the two sentences together into this: "Annual average NH3 mixing ratio increased from 2009 to 2017 by 50% and then decreased by 49% from 2017 to 2020."

Line 19: by 60%: compared to which level?

----- The "60%" reduction is relative to the initial concentration. We have revised the sentence: Thermodynamic modeling revealed the nonlinear response of SIAs to NH₃, with increased sensitivity when its concentration is reduced to 40% of the initial level.

Line 22: change to this: "Nevertheless, the increased mass proportion of ammonium salts in SIAs…"

Line 32: change "Research on…" to "Existing studies on…"

Line 35: change "debate" to "a debate"

Line 47: change "countries have conducted studies" to "studies have been conducted"

Line 58: change "explore" to "explored"

List of all changes made in the manuscript:

1) Line 12: change "their" to "its".

2) Lines 14-16: change "$NH_3$ mixing ratios initially increased and peaked in 2017 but subsequently decreased. From 2009 to 2017, $NH_3$ mixing ratios increased by 50%, while there was a decrease of 49% in 2020 compared to 2017." to "Annual average $NH_3$ mixing ratio increased from 2009 to 2017 by 50% and then decreased by 49% from 2017 to 2020."

3) Line 19: change "to $NH_3$ when its concentration decreases by 60%." to "when its concentration is reduced to 40% of the initial level."

4) Line 22: change "in the proportion (mass proportion concentration)" to "mass proportion".

5) Line 32: change "Research" to "Existing studies".

6) Line 35: change "debate" to "a debate".

7) Line 47: change "countries have conducted studies" to "studies have been conducted".

8) Line 58: change "explore" to "explored".

9) Line 61: change "aims" to "aimed".

10) Line 65: change "Additionally" to "Furthermore".

11) Line 79: change "concentrations," to "concentrations;".

12) Line 85: change "Teflon line and," to "Teflon line, and".

13) Line 137: change "mode" to "model".

14) Line 140: change "(unit is k)" to "(in Kelvin)".

15) Line 141: change "($SO_4^{2-}$、$Cl^-$ + HCl、$NO_3^-$ + $HNO_3$、$NH_4^+$ + $NH_3$、$Na^+$、$K^+$、$Ca^{2+}$ and $Mg^{2+}$)" to "($SO_4^{2-}$, $Cl^-$ + HCl, $NO_3^-$ + $HNO_3$, $NH_4^+$ + $NH_3$, $Na^+$, $K^+$, $Ca^{2+}$ and $Mg^{2+}$)".

16) Line 203-205: change "The emission inventory data obtained (Figure 1) indicate that from 2009 to 2014, the total $NH_3$ emissions in Beijing remained stable, peaking in 2012. After 2014, $NH_3$ emissions in Beijing rapidly decreased, declining by 25% from 2012 to 2017." to "The emission

inventory data (Figure 1) indicate that NH$_3$ emissions in Beijing remained stable from 2009 to 2014, peaking in 2012. After 2014, NH$_3$ emissions in Beijing rapidly decreased, declining by 25% from 2014 to 2017."

17) Line 282: change "Averaging minute-level data over one hour can smooth the effects the effect caused by variations in humidity" to "Averaging minute-level data over one hour can smooth the effects caused by variations in humidity".

18) Line 291: change "infer" to "suggest".

19) Line 322: change "investigates" to "investigated".

20) Line 333: change "relationship ion NH$_4^+$ in fine particulates and atmospheric NH$_3$" to "relationship of NH$_4^+$ in fine particulates to atmospheric NH$_3$".

21) Line 337: change "the highest" to "was the highest".

22) Line 341: change "which in turn exacerbated" to "exacerbating".

23) Line 356-357: change "As also suggested by Su et al. (2021), the acidic components" to "Su et al. (2021) also suggested that the acidic components".